# Risk factors, symptom reporting, healthcare-seeking behaviour and adherence to public health guidance: protocol for Virus Watch, a prospective community cohort study

Andrew Hayward,[1] Ellen Fragaszy,[2,3] Jana Kovar,[1] Vincent Nguyen,[1,2] Sarah Beale,[1,2] Thomas Byrne,[2] Anna Aryee,[2] Pia Hardelid [iD] ,[4] Linda Wijlaars [iD] ,[5,6] Wing Lam Erica Fong,[2] Cyril Geismar,[1,2] Parth Patel,[2] Madhumita Shrotri,[2] Annalan M D Navaratnam,[1,2] Eleni Nastouli,[5,7] Moira Spyer,[5,7] Ben Killingley,[8,9] Ingemar Cox,[10] Vasileios Lampos,[10] Rachel A McKendry,[11] Yunzhe Liu,[12] Tao Cheng,[12] Anne M Johnson,[13] Susan Michie,[14] Jo Gibbs,[13] Richard Gilson,[13] Alison Rodger,[13,15] Robert W Aldridge [iD] [2]

► Prepublication history and supplemental material for this paper is available online. To view these files, please visit the journal online (http://dx.doi.org/10.1136/bmjopen-2020-048042).

For numbered affiliations see end of article.

**Correspondence to**
Dr Andrew Hayward;
a.hayward@ucl.ac.uk

## ABSTRACT

**Introduction** The coronavirus (COVID-19) pandemic has caused significant global mortality and impacted lives around the world. Virus Watch aims to provide evidence on which public health approaches are most likely to be effective in reducing transmission and impact of the virus, and will investigate community incidence, symptom profiles and transmission of COVID-19 in relation to population movement and behaviours.

**Methods and analysis** Virus Watch is a household community cohort study of acute respiratory infections in England and Wales and will run from June 2020 to August 2021. The study aims to recruit 50 000 people, including 12 500 from minority ethnic backgrounds, for an online survey cohort and monthly antibody testing using home fingerprick test kits. Nested within this larger study will be a subcohort of 10 000 individuals, including 3000 people from minority ethnic backgrounds. This cohort of 10 000 people will have full blood serology taken between October 2020 and January 2021 and repeat serology between May 2021 and August 2021. Participants will also post self-administered nasal swabs for PCR assays of SARS-CoV-2 and will follow one of three different PCR testing schedules based on symptoms.

**Ethics and dissemination** This study has been approved by the Hampstead National Health Service (NHS) Health Research Authority Ethics Committee (ethics approval number 20/HRA/2320). We are monitoring participant queries and using these to refine methodology where necessary, and are providing summaries and policy briefings of our preliminary findings to inform public health action by working through our partnerships with our study advisory group, Public Health England, NHS and government scientific advisory panels.

## Strengths and limitations of this study

► Virus Watch is a large national household community cohort study of the occurrence of and risk factors for COVID-19 infection that aims to recruit 50 000 people, including 12 500 from minority ethnic backgrounds.

► Virus Watch is designed to estimate the incidence of PCR-confirmed COVID-19 in those with respiratory and non-respiratory presentations and the incidence of hospitalisation among PCR-confirmed COVID-19 cases.

► Virus Watch will measure effectiveness and impact of recommended COVID-19 control measures including testing, isolation, respiratory and hand hygiene measures, and social distancing on risk of respiratory infection.

► Only households with a lead householder able to speak English are able to take part in the study. Participant information sheets and consent forms are available in 9 languages but the study surveys are in English, limiting participation for non-English speaking households.

► Only households of up to six people were eligible for inclusion and they are also required to have access to an internet connection. These restrictions will limit the generalisability to large or multigenerational households, and those without access to the internet.

## INTRODUCTION

The COVID-19 pandemic has caused millions of deaths and impacted lives around the world with the closure of schools, workplaces and limitations on freedom of movement. Vaccines and effective scalable treatments for COVID-19 have been developed and while these are rolled out across England and Wales we will need to rely on other measures to stop the spread of COVID-19. We will also require

studies to examine their long-term effectiveness as they are implemented across England and Wales.

Governments, including those of the UK devolved nations, are adopting a wide range of control measures to limit the spread of infection. These include isolation of people with COVID-19 symptoms and their household contacts, widespread testing and contact tracing, digital contact tracing using mobile phone apps, broad social distancing measures and local control measures. Environmental cleaning, hand hygiene and face mask use are also advised.

Much of our current knowledge of COVID-19 comes from observations at the more severe end of the disease spectrum, in hospitalised patients and individuals who die having tested positive for the disease.[1–3] Although large-scale studies of prevalence of PCR positive infection and seroprevalence have been established, there is currently limited information on symptom profiles through the course of illness in non-hospitalised populations, children, social and behavioural risk factors for infection, strength and duration of immunity, household and community transmission risk, and population behaviours during periods of wellness and illness (including social contacts, use of public spaces, testing behaviours, isolation, mask use, hand and respiratory hygiene). This information can only be gathered accurately through prospective large-scale community cohorts. Our experience of the Medical Research Council (MRC)/Wellcome Flu Watch study[4 5] and the Economic and Social Research Council (ESRC) Bug Watch[6] study has allowed us to rapidly establish a national household cohort study of 50 000 individuals.

Virus Watch aims to provide evidence on which public health approaches are most likely to be effective in reducing the spread and impact of the virus and will investigate community incidence, symptom profiles and transmission of COVID-19 in relation to population movement and behaviour.

## METHODS AND ANALYSIS
### Study design and setting

Virus Watch is a household community cohort study of acute respiratory infections in England and Wales covering the second and potential subsequent waves of the COVID-19 pandemic. The study period will be from 1 June 2020 to 31 August 2021. The study aims to recruit 50 000 individuals, including 12 500 from minority ethnic backgrounds for an online survey cohort (study 1). Nested within this larger study will be a subcohort of 10 000 individuals (study 2), including 3000 people from minority ethnic backgrounds. Participants in this laboratory subcohort will be selected based on their geographical distance from one of our blood-taking clinics; either a 10 km radius from a clinic in cities or a 20 km radius in rural areas. Participants will be balanced to be representative of the UK population for sex, age and region. Figure 1 provides an overview of the study design.

Households self-select into the study if they live in England or Wales and all members of a household need to consent to take part in the study to meet our inclusion criteria (online supplemental appendix 1). Households need to have an internet connection on a phone, tablet or computer, email, and at least one adult household member that can read English.A household is defined as one or more people (not necessarily related) whose usual residence (4 days/week or more) is at the same address. These householders share cooking facilities, a living room or sitting room or dining area.

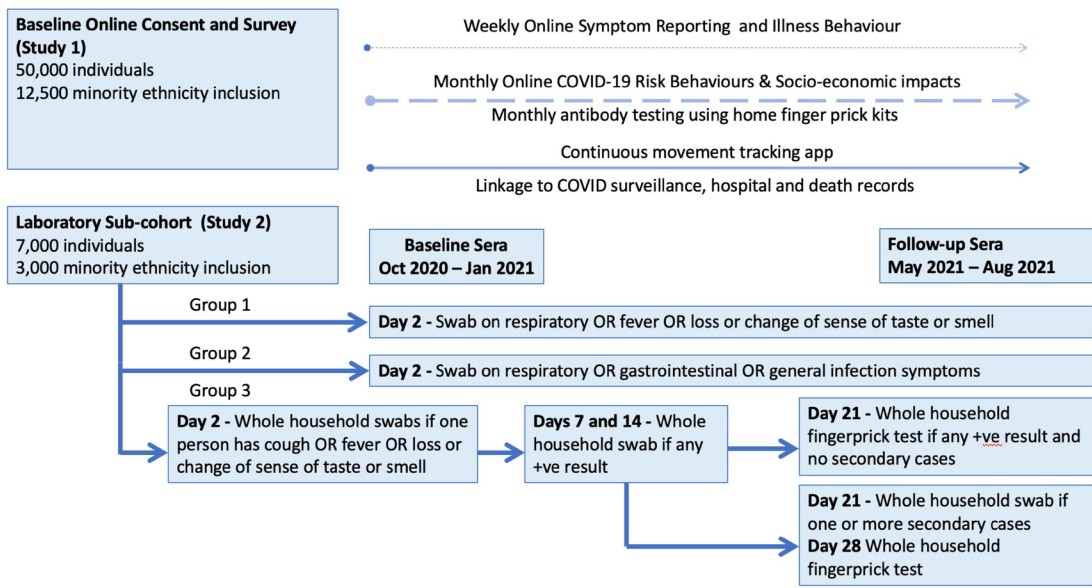

**Figure 1** Overview of cohort recruitment, PCR swabbing schedules and data collection for the Virus Watch household community cohort study.

## Primary outcomes

### Study 1: online survey cohort

1. Incidence of respiratory infection symptoms, including COVID-19 disease case definitions.
2. Effectiveness and impact of recommended COVID-19 control measures including testing, isolation, social distancing, respiratory and hand hygiene measures on risk of respiratory infection.
3. Frequency of adherence to public health recommendations for these control measures.
4. Proportion of community infections that result in hospital admissions and death.
5. Vaccine effectiveness against asymptomatic and symptomatic infections.

### Study 2: laboratory testing subcohort

1. Incidence of PCR-confirmed COVID-19.
2. Incidence of PCR-confirmed COVID-19 in those with non-respiratory presentations.
3. Incidence of hospitalisation among PCR-confirmed COVID-19 cases.
4. Proportion of individuals with SARS-CoV-2 antibodies acquired through natural infection to pandemic coronavirus.
5. Proportion of individuals with cross-reacting antibodies to seasonal coronaviruses acquiring (or not) SARS-CoV-2.
6. Household secondary attack rates.
7. Protective effect of antibodies on infection and reinfection as well as the severity and spectrum of presentation.

## Recruitment

We will use the Royal Mail Post Office Address File to generate a list of residential address lists from which households can be sampled and sent Virus Watch recruitment postcards to. The proposed initial sample design is a single-stage stratified probability sample where implicit stratification is employed to benefit from the precision gains that stratified sampling can bring. Within each region, residential addresses are sorted by (A) quintiles of Index of Multiple Deprivation 2019, (B) within quintiles by local authorities, (C) postcodes, and (D) address. We will perform this in the nine government office regions of England as well as Wales (10 study regions in total).

We will assess recruitment rates and the representativeness of this initial sample following the mail out of 50000 postcards. If recruitment is lower than expected or under-representative of the national population, we will redesign our recruitment campaign to include a range of methods in order to build the cohort. This mixed recruitment strategy will be flexible and use a variety of methods including social media, study leaflet drops, text messaging, personalised letters and incentives. Social media adverts will be used to inform individuals about the study and direct them to our website, http://ucl-virus-watch.net/, where they can read the participant information sheets and consent to taking part. Digital invitations will also be created for sharing via WhatsApp. Text messages and postal letters inviting patients from their general practitioner (GP) clinics will be organised via Local Clinical Research Networks.[7] We will also work with trusted community partners and religious organisations to promote recruitment into the study.

In order for a household to be enrolled, they will require an internet connection (Wi-Fi, fixed or on a mobile phone), email address, and all household members must agree to take part. Households will nominate a lead householder who will submit study questionnaires. The lead householder will need to be able to read English to support other household members in survey completion. A household is defined as one or more people (not necessarily related) whose usual residence (4 days/week or more) is at the same address. These householders share cooking facilities, and may share a living room or sitting room or dining area if available. Households with more than six members will not be eligible for the study—this criterion was set due to limitations of the Research Electronic Data Capture (REDCap) survey infrastructure which did not function correctly when attempting to work with household sizes of greater than six during our pilot testing of the survey.

Virus Watch is powered for our primary aims in study 2 and the estimation of population-level symptomatic COVID-19 attack rate over time. Recruiting a cohort that is representative of the population is time consuming as it requires an initial invitation into a study followed by multiple follow-up contacts encouraging invited individuals to register. Given the urgency of the public health situation to roll out our study as quickly as possible, we chose a different approach whereby we recruit a large cohort of 50000 individuals and from within that cohort we select a subsample for the testing cohort (subcohort 1) which is representative of the population in terms of age, sex, ethnicity, region, household size and proportion of households with children. The larger cohort will be important in assessing rates and predictors of less frequent outcomes such as hospitalisation and death. Given recent information of marked ethnicity differences in mortality rates from COVID-19, we also chose to recruit an ethnicity sample designed to be sufficiently large to provide early indicators of whether these differential mortality rates are due to differences in disease incidence or in differences in severity or both.

## Power analysis

The testing subcohort is powered for accurate weekly age-specific disease incidence rates to be measured assuming 20%–30% clinical attack rate over 18 weeks. With a clinical attack rate of 30% of whom 20% need hospitalisation and 0.5% die, we expect the following number of outcome events in our testing cohort of 10000 individuals in study 2: 3000 COVID-19 illnesses, 600 hospitalised cases and 15 deaths. At 1month into the outbreak we would be able to detect a 1.7-fold greater risk of disease in a population subgroup that constitutes one-fifth of the population, and

by 2 months the detectable relative risk would be only 1.2. At 1 month we could detect a 4% hospital admission rate among cases with 95% CI of 0.5 to 6.8, and by 2 months the CIs would narrow to 3.1 to 4.1. We have used estimates of the expected number of events over time to provide an indication of the fact that the cohort is sufficiently large to provide valuable information through the course of the pandemic. Sample size calculations have been informed by a realistic assessment of what we can achieve based on our previous experience.[4 6] For the serology cohort of 3000 people from minority ethnic backgrounds we assume a modest design effect due to household and geographical clustering, and 500 participants for six different minority ethnic backgrounds would enable the measurement of a cumulative incidence of 10% with 95% CIs of 3% by each minority ethnic group.

## Participant materials and incentives

Participant information sheets will be held on our study website (these along with consent forms were translated into six languages and a further three languages were added from December 2020). In order to participate, the whole household must take part. Each adult participant will need to read through study information, and provide online informed consent for themselves and any children they are legally responsible for. Children aged 6–9 and 10–15 years, respectively, will also be asked to read through age-specific study participant information sheets and provide online informed assent. For children aged 5 and under, parents/guardians will consent on their behalf. Informed consent data will be securely stored in the University College London (UCL) Data Safe Haven which has been certified to the ISO27001 information security standard and conforms to the National Health Service (NHS) Digital's Data Security and Protection Toolkit. Local study teams will reconsent participants face to face, prior to undertaking blood sampling, and adult participants in study 2 will be offered a £10 voucher to reimburse travel costs, if required. From February 2021, invitation letters sent by GP clinics will include a £20 voucher for households who agree to take part in the study.

## Data collection and follow-up
### Study 1: online survey cohort

The online survey cohort will collect data and follow up participants through six different sources. Survey data will be collected using REDCap electronic data capture tools hosted on the UCL Data Safe Haven (online supplemental appendices 2–4).[8] REDCap is a secure, web-based application for research studies. The UCL Data Safe Haven provides a technical solution for storing, handling and analysing identifiable data. It has been certified to the ISO27001 information security standard and conforms to NHS Digital's Data Security and Protection Toolkit .

1. Baseline survey. The lead householder will be asked to complete an online baseline survey for each member of their household. Information collected includes:

demographics, occupation, income, ethnicity, country of birth, year of entry to UK, chronic medical conditions, medications, pregnancy status, vaccines, mode of transport to work, any previous contact with someone with COVID-19, previous symptoms of COVID-19-like illness and infection prevention behaviours such as social distancing and hand hygiene.

2. Illness surveys. Participants will be followed up weekly via an email with a link to an illness survey. This is a weekly survey of the presence or absence of symptoms that could indicate COVID-19 disease including respiratory, general infection or gastrointestinal symptoms. During illness, prospective daily symptom recording, quality of life, health-seeking behaviour (NHS 111, GP in person, GP by phone, accident and emergency, pharmacy, hospital), treatments and NHS investigations will be recorded. This survey will also include any respiratory and hand hygiene measures, self-isolation, activities and social contact, travel and face mask use. Questions around behavioural interventions, such as mask wearing and social distancing, aim to reflect the context and frequency/degree to which behaviours are practised according to governmental and public health guidelines and relevant scientific literature. The survey includes questions to the household on activities undertaken in the week prior to symptom onset. The weekly survey will also be used to capture test results received from outside the study and requests to self-isolate, for example, via the UK Test-Trace-Isolate system. The weekly survey will also ask about participants' COVID-19 vaccination uptake, including their date of vaccination, dose (ie, first or second) and which vaccine was administered.

3. Monthly surveys. A number of questions will be asked every month. The monthly surveys also provide flexibility to ask additional questions (eg, behavioural changes) to reflect any new government directives on social distancing, testing, contact tracing and vaccine delivery. Core questions will also allow us to follow up reasons for any non-response in a given month (eg, because of illness, hospitalisation or holiday). We will also ask about online health information seeking, social distancing, including recent (week before) contacts, activities, places visited and hand and respiratory hygiene. As with the weekly questionnaire, questions around behavioural practices will reflect governmental and public health guidelines and the scientific literature; monthly questionnaires will also investigate barriers and enablers to health-related behaviours using purpose-developed questionnaires based on the Capability, Opportunity, Motivation, Behaviour model. We will also ask about finances, employment and mental health to see how the COVID-19 response is affecting participants' well-being and ability to work. We will ask about access to healthcare for non-COVID-19 health problems to explore the indirect health impacts of the pandemic. We will ask about any COVID-19 PCR or antibody test results performed outside the study and not

already reported through baseline surveys. We will ask about influenza vaccine uptake and COVID-19 vaccination intentions.

4. Data linkage. NHS Digital will undertake quarterly data linkage between cohort 1 and Hospital Episode Statistics (HES), which includes admitted patient and critical care episodes, outpatient department bookings and emergency care contacts. This linkage will also include Office for National Statistics mortality data, COVID-19 vaccination records and virology testing data routinely collected by Public Health England (PHE), Public Health Wales and the Department of Health and Social Care through 'Pillar 1' (testing in hospital patients and health and care workers) and 'Pillar 2' (community testing). These data sources will be linked to the cohort using name, NHS numbers, dates of birth and postal address. Identifying variables will be removed before the linked data are transferred back to UCL for analysis. These data linkages will continue for up to 5 years after the end of the study as we anticipate COVID-19 will become a recurring winter infection and we wish to understand its impact on health services in subsequent years. These linkage studies will identify any participants who have been admitted to hospital or died due to causes that could be directly or indirectly linked to the COVID-19 pandemic. Indirect causes include those related to limitations in healthcare access during the pandemic. Reductions in the use of routine health services will also be monitored via linkage to HES data.

5. Geolocation tracking. All adult participants will be asked about optional consent to use a secure geolocation tracking app (Tracker for ArcGIS) installed on their mobile phone for the duration of the study.

6. Monthly antibody testing using home fingerprick kits. Adults aged 18 years and over enrolled in the online survey cohort, with the exception of those in laboratory testing subcohort group 3, will be offered monthly antibody testing starting February 2021 and continuing until the end of the study, using home fingerprick kits for self-collection of capillary blood samples. Those aged under 18 and living with adults enrolled in monthly antibody testing will continue completing online surveys. Monthly antibody testing (February–August 2021) will use Conformitè Eurōpēenne (CE)-marked at-home fingerprick kits designed to collect small-volume (400–600 µL) capillary blood samples. Samples are self-collected by adult participants and returned to a United Kindom Accreditation Service (UKAS)-accredited laboratory via prepaid post, where they will be tested for anti-nucleocapsid and anti-spike antibodies using validated electrochemiluminescence immunoassays.

## Study 2: laboratory testing subcohort
All participants agreeing to take part in the main cohort (study 1) will be asked to provide consent to be contacted and invited to participate in one of the three laboratory testing subgroups. This will enable a cohort of 10 000 individuals selected from the main cohort of 50 000 individuals to be maximally representative of the population of England and Wales. All participants taking part in study 2 will be asked to use the national test, trace and isolation system in addition to providing samples as part of Virus Watch.

Study 2 will consist of three groups that will follow different schedules of antibody testing and nasal/throat swabs for PCR testing.

### Group 1 (n=7000)
With data from this group we aim to identify infection in those with a wide range of respiratory symptoms. Participants will be asked to submit a nose/throat swab if they experience 2 consecutive days of: fever (>37.8°C), feeling feverish, or new persistent cough, or loss or altered sense of smell or taste (COVID-19 suspected case definition), or shortness of breath, or ear pain or change in hearing, or sore throat, or sneezing, or blocked nose, or runny nose, or wheeze or sinus pain or congestion (other respiratory manifestations).

### Group 2 (n=1000)
This group aims to identify the importance of non-respiratory presentations. Participants will be asked to submit a self-taken nasal/throat swab for PCR identification of COVID-19 and other respiratory viruses if:
► Either 2 consecutive days of respiratory symptoms (eg, cough, runny nose, sneezing, shortness of breath, sore throat, blocked nose, sinus pain or congestion, ear pain or change in hearing, wheezing, loss of or altered sense of taste or sense of smell).
► Or 2 consecutive days of gastrointestinal symptoms (eg, diarrhoea/loose stools, abdominal pain, nausea or vomiting, loss of appetite).
► Or 2 consecutive days of general infection symptoms (eg, feeling feverish, having a high temperature, feelings of severe unexplained tiredness, generalised muscle or joint aches).

### Group 3 (n=2000)
This group aims to identify the extent of household transmission. Participants will be asked to submit a nose/throat swab if they experience 2 consecutive days of cough or fever or loss of sense of taste or smell. Household contacts of the index case will also be asked to submit a swab on the same day whether or not they have symptoms.

If any of the swabs indicate SARS-CoV-2 infection, all household members will be asked to repeat the swab on day 7 and day 14. If there are no new SARS-CoV-2 cases in the household arising from swabs on days 7 and 14 (assumed secondary cases) then all household members will be asked to undertake a home fingerprick antibody test on day 21. If there is one or more secondary cases in the household then the entire household will be asked to take an additional swab on day 21 and then undertake the fingerprick antibody tests on day 28.

### End of follow-up

Online participant follow-up will end in August 2021 for households enrolled in monthly antibody testing, and in May 2021 for others, although depending on the progression of COVID-19, we may ask participants to continue in the study for longer. Participants will be sent an exit survey. Participants will be contacted to arrange a second blood sample collection from April 2021. Follow-up through data linkage with HES, COVID-19 vaccination records and mortality data will continue for 5 years after the end of the study.

### Laboratory testing

#### Antibody testing

Study 2 will be using two different types of antibody tests. First, full blood serology will be taken between October 2020 and January 2021. We will use experienced healthcare professionals, including research nurses from the National Institute for Health Research Clinical Research Networks.[7] Depending on local circumstances, visits to participants' homes to take blood may also be arranged. Children aged 15 years or less can opt out of having their blood taken but will be offered a fingerprick antibody test conducted by a healthcare worker instead. All participants from laboratory group 3 will additionally be offered a fingerprick antibody test at the same time as blood taking. From April 2021 until July 2021, we will invite all participants back for full blood tests or, for children who do not wish to have a full bleed, healthcare worker-delivered fingerprick-based antibody tests.

Families of children who have not been able to attend for a blood test, or for a healthcare worker-delivered fingerprick antibody test, will be provided with postal kits to perform these at home. We also plan to use fingerprick antibody testing where local clinics are no longer able to undertake full blood tests due to COVID-19 travel restrictions. Extremely clinically vulnerable participants will be sent home fingerprick tests instead of being asked to provide a serological sample.

#### Virus detection

Participants will post swab samples for PCR assays of SARS-CoV-2, and subsequent testing for influenza virus, seasonal coronavirus, rhinovirus and respiratory syncytial virus. When SARS-CoV-2 is identified we will also undertake whole-genome sequencing of the virus. Samples for COVID-19 diagnostics will be handled and processed according to the NHS and UCL guidance on sample handling during the COVID-19 pandemic.

COVID-19 PCR and serology results will be returned to participants via email message systems. These messages will include links to official support, information and advice from NHS and PHE as well as advice on how to interpret results based on current evidence. In laboratory group 3, where positive test results will trigger further testing of the household, the results email will also include details explaining the additional testing requests. We will be not asking for inconclusive PCR results to be repeated

### Statistical analysis

Our primary analyses during the winter 2020/2021 season will focus on estimating age-specific weekly rates of symptoms and risk factors for PCR-confirmed COVID-19 illness and hospitalisation. For these analyses we will use Poisson regression models that account for clustering by household using robust SEs and we will explore the use of stratification or weighting of the sample by age and region as necessary to give nationally representative estimates. Weekly rates will be expressed per 100 000 person-weeks for ease of comparison with national surveillance data.

We will examine the proportion of the population infected during the first wave (eg, February–September 2020) and second and potential future pandemic waves. We will estimate the percentage of the population infected by calculating age and wave-specific rates of serological infection and PCR-confirmed disease per 100 person-seasons using Poisson regression with robust SEs to account for household-level clustering. A person-season will be defined by the epidemic curve in the cohort and therefore rates will account for differential follow-up time during each epidemic peak. In these analyses we will examine risk factors for infection, disease, disease severity and disease transmission.

We will estimate the proportion of serologically confirmed SARS-CoV-2 infections leading to symptomatic disease. First, we will calculate age-adjusted attributable rates of illness due to infection (subtracting rates of respiratory illness in non-seroconverters from those in seroconverters). Second, we will measure the proportion of seroconverters with PCR-confirmed COVID-19. Analyses plans will be developed prior to conducting all analyses.

We will estimate vaccine effectiveness against asymptomatic SARS-CoV-2 infections and against symptomatic COVID-19 using anti-nucleocapsid seroconversion, positive PCR testing and self-reported symptom data. We will use both time-to-event and test-negative analytical frameworks. Using quantitative antibody data, we will assess the dynamics of anti-spike antibodies over time and the relationship between antibody titres and the risk of infection.

While the study is being conducted, we will produce early, preliminary results and analyses for participants, the general public, government scientific advisory groups and policymakers in order to inform the public health response to the pandemic. These analyses will be reactive to the epidemiological circumstances and are therefore not defined in this protocol.

### Modelling

We will build on our experience of working with PHE, Google and Microsoft to use anonymous national or subnational aggregate web search engine data[9 10] to monitor the spreading of the disease. We will use our study data as ground truth to train real-time disease prevalence estimation algorithms. We will annotate Global Positioning System tracking data into standard categories including time at work and home, social venues, supermarkets,

hospitals, GPs and transport mode for incorporation in classical epidemiological analyses. Integrating the linked survey data, we will develop a predictive spatiotemporal transmission model to investigate the impact of various social distancing strategies.

## Missing data

We have several strategies that attempt to address the issue of missing data. First, we have sought to minimise the amount and impact of missing data for key outcomes and exposures through the study design. For example, for a number of our primary outcomes (PCR-confirmed illness, hospitalisation and death) and exposures (vaccination) we collect data both as self-reported and through data linkage with the relevant national data sets and registries. Second, we sought to minimise missing serological and Virus Watch specific swabbing outcomes in adults by making willingness to provide relevant specimens a prerequisite to study registration. Third, we know from our experience of previous community cohort studies of acute infections (Flu Watch[4] and Bug Watch[6]) that response to weekly surveys (where our symptom data are collected) is high at around 75%, which we believe is achieved by keeping these weekly data collections simple and quick to complete. We have aimed to replicate this approach in Virus Watch. Fourth, for important missing baseline demographic data (eg, age and sex) we have created follow-up surveys to try and collect missing data at a later point in time. Fifth, where necessary, we will address missing data in our analyses and use multiple imputation methods if appropriate.

## Patient and public involvement

Due to the urgent nature of this study, we did not involve participants in its original design. We have previously conducted patient and public involvement to support similar community cohort studies of acute infections using similar methodologies. We have engaged the Young Persons Advisory Group for research at Great Ormond Street Hospital to provide feedback on our Children's Participant Information Sheets. We have worked with the Race Equality Foundation and Doctors of the World in advising on the inclusion of people from minority ethnic backgrounds in Virus Watch and have set up an advisory group to inform the ongoing design and dissemination of health equity aspects of Virus Watch. They were not asked to assess the burden of the intervention and time required to participate in the research due to the urgent nature of setting the study up. This advisory group (consisting of lay members of the public, community leaders, charities and policy organisations who will be reimbursed for their time) will guide our health equity analyses and steer us on their implications for people, communities and policy. The advisory group will also help us prioritise what information and results to share, when and in what format.

## ETHICS AND DISSEMINATION

This is a national study that has been approved by the Hampstead NHS Health Research Authority Ethics Committee (ethics approval number 20/HRA/2320). The study is compliant with the requirements of General Data Protection Regulation (2016/679) and the Data Protection Act (2018). All investigators and study site staff will comply with the requirements of the General Data Protection Regulation (2016/679) with regard to the collection, storage, processing and disclosure of personal information, and will uphold the Act's core principles.

We will provide opportunities for survey participants to comment on survey methodology in the first monthly survey and consider revisions based on this. We are also monitoring participant queries through our study email address and using these to refine methodology where necessary.

## Data sharing and access

We aim to share aggregate data from this project on our website and via a 'Findings so far' section on our website—https://ucl-virus-watch.net/. We will also be sharing individual record-level data with personal identifiers removed on a research data-sharing service such as the Office for National Statistics Secure Research Service.[11] In sharing the data we will work within the principles set out in the UK Research and Innovation (UKRI) guidance on best practice in the management of research data.[12] Access to use of the data while research is being conducted will be managed by the chief investigators (AH and RWA) in accordance with the principles set out in the UKRI guidance on best practice in the management of research data. It is the intention that the data arising from this research will initially be collected, cleaned and validated by the UCL research team and once this has been completed will be shared for wider use. We aim to make subsets of the data more rapidly available both on our study website and via the public-facing dashboard during the ongoing phase of data collection. In line with Principle 5 of the UKRI guidance on best practice in the management of research data, we plan to release data in batches as they become available or as updated results are published. Individual record data linked using NHS Digital will not be shared, only aggregated results. HES and mortality data may be obtained from a third party and are not publicly available. These data are owned by a third party and can be accessed by researchers applying to the Health and Social Care Information Centre for England. We will put analysis code on publicly available repositories to enable their reuse.

**Author affiliations**
[1]Institute of Epidemiology and Health Care, University College London, London, UK
[2]Centre for Public Health Data Science, Institute of Health Informatics, University College London, London, UK
[3]Department of Infectious Disease Epidemiology, LSHTM, London, UK
[4]Centre for Paediatric Epidemiology and Biostatistics, Institute of Child Health, University College London, London, UK
[5]Population, Policy and Practice, University College London, London, UK

[6]Primary Care and Population Health, University College London, London, UK
[7]Francis Crick Institute, London, UK
[8]University of Nottingham School of Medicine, Nottingham, UK
[9]University College London Hospital, London, UK
[10]Department of Computer Science, University College London, London, UK
[11]London Centre for Nanotechnology and Division of Medicine, University College London, London, UK
[12]SpaceTimeLab, Department of Civil, Environmental and Geomatic Engineering, University College London, London, UK
[13]Institute for Global Health, University College London, London, UK
[14]Centre for Behaviour Change, University College London, London, UK
[15]Royal Free London NHS Foundation Trust, London, UK

**Contributors** Conceptualisation: AH, EF, JK, PH, EN, BK, IC, VL, RAM, TC, AMJ, SM, JG, RG, AR, RWA. Investigation, methodology: all authors. Project administration: AH, EF, JK, VN, SB, TB, AA, PH, LW, WLEF, CG, PP, MSh, AMDN, EN, MSp, RWA. Writing–original draft preparation: all authors. Software: VN, TB, SB, RWA. Resources: AH, EF, JK, PH, EN, BK, IC, VL, RAM, TC, YL, AMJ, SM, JG, RG, AR, RWA. Writing–review and editing: all authors.

**Funding** The research costs for the study have been supported by the MRC Grant Ref: MC_PC 19070 awarded to UCL on 30 March 2020 and MRC Grant Ref: MR/V028375/1 awarded on 17 August 2020. The study also received $15 000 of Facebook advertising credit to support a pilot social media recruitment campaign on 18 August 2020.

**Competing interests** AH serves on the UK New and Emerging Respiratory Virus Threats Advisory Group. AMJ was a governor of Wellcome Trust from 2011 to 2018 and is chair of the Committee for Strategic Coordination for Health of the Public Research.

**Patient consent for publication** Not required.

**Provenance and peer review** Not commissioned; externally peer reviewed.

**ORCID iDs**
Pia Hardelid http://orcid.org/0000-0002-0154-1306
Linda Wijlaars http://orcid.org/0000-0003-1222-2922
Robert W Aldridge http://orcid.org/0000-0003-0542-0816

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
