## [Reviewer comments · BMJ Open]

ARTICLE DETAILS

TITLE (PROVISIONAL)	Risk factors, symptom reporting, healthcare-seeking behaviour and adherence to public health guidance: protocol for Virus Watch, a prospective community cohort study
AUTHORS	Hayward, Andrew; Fragaszy, Ellen; Kovar, Jana; Nguyen, Vincent; Beale, Sarah; Byrne, Thomas; Aryee, Anna; Hardelid, Pia; Wijlaars, Linda; Fong, Wing Lam Erica; Geismar, Cyril; Patel, Parth; Shrotri, Madhumita; Navaratnam, Annalan M D; Nastouli, Eleni; Spyer, Moira; Killingley, Ben; Cox, Ingemar; Lamos, Vasileios; McKendry, Rachel; Liu, Yunzhe; Cheng, Tao; Johnson, Anne; Michie, Susan; Gibbs, Jo; Gilson, Richard; Rodger, Alison; Aldridge, Robert

VERSION 1 – REVIEW

REVIEWER	Mangal, Tara Imperial College London, Infectious Disease Epidemiology
REVIEW RETURNED	20-Jan-2021

GENERAL COMMENTS	This is a well-considered trial of particular importance in the current situation. The outcomes of the trial are commendably broad and will have direct implications for management of risk and interventions in the coming months. The trial is well underway and I have no concerns about the methodology, just a few small comments / clarifying questions that I would like to make. Data on chronic conditions are collected (presumably as risk factors) but it would be useful to know what the intentions are regarding these data. Which conditions will be included in the analyses; will any infectious diseases also be analysed as potential risk factors for disease severity / mortality, will the stage of chronic disease be considered (late stage vs early stage cancers or CVD)? How about obesity? Are there any intentions to stratify any of the analyses by virus strain? I understand that perhaps when the protocol was written, there was only one dominant circulating strain but as the protocol seems to be reactive to the changing epidemiology, it would be incredibly useful to include these data if possible. Vaccination status is collected, hopefully the vaccine type along with the dates of vaccination are also recorded. Could vaccine efficacy also be estimated using these data? Duration of protection perhaps could also be estimated as there is a 5-year follow-up planned.
---

	Certain behavioural interventions are hard to quantify, such as the extent of proper use of masks. How is this defined? One limitation is that only households with up to 6 persons can be included due to the REDCap survey infrastructure. This means that large multigenerational households will automatically be excluded and we know that this is an important risk factor for infection, particularly in BAME communities. The 5 year follow-up for indirect consequences of COVID-19 is particularly interesting, giving insights into how disruptions to the health system could impact longer-term health. This could potentially be one of the main aims as this study is uniquely placed to address this question. Given that we know that there is a significantly increased risk in both infection and disease severity in care home residents, has this been considered in either the recruitment or analysis plan? If self-administered nasal/throat swabs return inconclusive results (always a risk for example when trying to test children - notoriously difficult!), are follow-up tests offered? Are data collected on whether children in the household are currently in school when samples / surveys are performed? This could be really useful for assessing the impact of school closures. Has the trial been registered? I couldn't find the details.
--	--

REVIEWER	Gwon, Yeongjin University of Nebraska Medical Center, Biostatistics
REVIEW RETURNED	30-Jan-2021

GENERAL COMMENTS	Risk factors, symptom reporting, healthcare-seeking behavior and adherence to public health guidance: protocol for Virus Watch, a prospective community cohort study Overall comment: The current manuscript presents a study protocol to investigate incidence, symptom profiles, and transmission of COVID-19 in connection with population movement and behaviors at community level in the UK. The strength and limitations were raised by the team as indicated in Page 7. Overall, the manuscript is well written and organized, however, I suggest a number of comments to improve and make the manuscript more accessible. Specific comment:  • Study design and setting  o Can the team justify a scientific justification why they recruit 42,500 individuals including 12,500 minority backgrounds group? To estimate the number participants in any study, the team need to provide a specific information how they compute this estimate. o Figure 1 simply presents an overview of cohort recruitment, and it has not directly associated with study design. Probably, the team need to revise the title of Figure 1. • Recruitment  o To be enrolled in the study, participants require an internet connection using computer or mobile device. I understand this strategy will be reasonable and make the study accelerate under current circumstances we face now. However, this inclusion criteria may not be able to reach out a population who live in urban
---

	area or who has lower income, assuming that more likely such people are not accessible to internet connection. Thus, the study samples may not well represent the population the team is targeting. Can the team clarify this issue? Or any other strategy to have them in the study?  o I would think that the team needs to make a table for inclusion criteria and then put it in Appendix. o It seems that the team already had an estimated clinical attack rate and its related quantities based on the previous data or their best knowledge. If it is, I encourage the team to write the paragraph (page 12) in the separate section, "Power Analysis". This should be an independent section to recruitment. • Statistical Analysis o There is no clear analysis plan. What do the authors mean by using "appropriate regression models?" This section is an overview of the entire analysis in this study. It would be better for the readers to understand what analyses (names of analyses) will be used for in this paragraph. o On Study 1, the data will be collected from the online survey. Due to the nature of survey study, the team will face lots of missing values, eventually. Can the team clarify and include their strategies how to deal with such missing values (primary outcomes and predictors including demographic information) in the statistical analysis framework? This is very important issue in data collection from survey, and based on the study protocol, participants are being asked multiple times (every month) for different questions. So, this should be incorporated or at least mentioned in statistical analysis plan. In addition, I am also concerned how the team would check the quality of the data. • Modelling o I am confused upon reading this section, honestly. My understanding is that the team is trying to develop a new predictive modeling for a better prediction on different social distancing strategies. As the team described, they will be using the study data to train real-time disease prevalence estimation algorithms (need to cite relevant literature if needed), which indicates that the team needs to perform a machine learning approach. In this sense, it will be more appropriate to use "develop a predictive spatio-temporal transmission models" rather than "develop multi-level spatio-temporal transmission models". I would like for the team to clarify this.
--	---

REVIEWER	Thome , Beatriz Universidade Federal de São Paulo, Medicina Preventiva
REVIEW RETURNED	05-Feb-2021

GENERAL COMMENTS	Thanks for the opportunity to review. Overall comment: Major strength of the study is to take advantage of an already existing community cohort. However I believe the objectives are ambitious, and the methods are not clear enough to understand if the objectives can be achieved through them, in particular objective 3 - "Virus Watch will measure effectiveness and impact of recommended COVID-19 control measures including testing, isolation, social distancing, respiratory and hand hygiene measures on risk of respiratory infection." Exclusion criteria seem to determine a major selection bias, that is not discussed in the limitations.
---

	Finally, this protocol will need to be significantly amended to encompass effect of vaccination among the control measures. Again, for this objective the methods are not clearly described. A few specific comments below: Introduction In light of vaccine roll out, the introduction will need considerable review. Also, will effectiveness of vaccines be included as one of the control measures? Study primary outcomes: will effectiveness of vaccine deployment be evaluated? If not, what is the plan to tease out the effect of the other measure, given vaccine roll out? Recruitment On "Households with more than six members will not be eligible for the study - this criteria was set due to limitations of the REDCap survey infrastructure which did not function correctly when attempting to work with household sizes of greater than six during our pilot testing of the survey.", I see a clear selection bias issue here. I wonder if study team would consider using a different software for the study instead of having this limitation as an exclusion criteria. Similar for the need to have internet and internet literacy to be included. Do authors have an idea of what proportion of the community stays out of the study with these exclusion criteria? Data collection and follow up "6) Home antibody finger prick tests. 5000 members of the online cohort who are not part of the Laboratory testing sub-cohort (including 2500 minority ethnic and 2500 White British people) will be offered home finger prick antibody testing kits as soon as available after the first wave of the pandemic and after the second wave of the pandemic." - can the authors provide more information on tests to be deployed? And what defines "wave"? What if epidemiology changes and there is no clear "wave"? Would suggest defining time points for serial testing rather. Great to see the "Patient and public involvement" section
--	---

VERSION 1 – AUTHOR RESPONSE

Reviewer comment 1-1:

This is a well-considered trial of particular importance in the current situation.

The outcomes of the trial are commendably broad and will have direct implications for management of risk and interventions in the coming months. The trial is well underway and I have no concerns about the methodology, just a few small comments / clarifying questions that I would like to make.

Author response 1-1:

Thanks very much for this helpful review - we appreciate the time you have taken to carefully read and provide constructive feedback on the article.

Reviewer comment 1-2:

Data on chronic conditions are collected (presumably as risk factors) but it would be useful to know what the intentions are regarding these data. Which conditions will be included in the analyses; will any infectious diseases also be analysed as potential risk factors for disease severity / mortality, will the stage of chronic disease be considered (late stage vs early stage cancers or CVD)? How about obesity?

Author response 1-2:

We now include all surveys as supplementary material. The following conditions are collected at baseline:

- Asthma
- Arthritis
- Congestive heart failure
- Coronary heart disease
- Angina
- Heart attack or myocardial infarction
- Stroke
- Emphysema
- Chronic bronchitis
- COPD (Chronic Obstructive Pulmonary Disease)
- Cystic fibrosis
- Hypothyroidism or an under-active thyroid
- Any kind of liver condition
- Cancer or malignancy
- Insulin treated diabetes
- Other diabetes
- Epilepsy
- High blood pressure/hypertension
- An emotional, nervous or psychiatric problem
- Multiple Sclerosis
- HIV
- Chronic kidney disease
- Conditions affecting the brain and nerves, such as Parkinson's disease, motor neurone disease, multiple sclerosis (MS), a learning disability or cerebral palsy
- Problems with your spleen or you've had your spleen removed

- Sickle cell disease
- Other long standing/chronic condition

If a participant states they have cancer, we specifically ask about the specific type of cancer with the following options:

- Bowel/colorectal
- Lung
- Breast
- Prostate
- Liver
- Skin cancer or melanoma
- Blood or bone marrow cancer, such as leukaemia
- Other

During the baseline data collection, participants are asked about their weight and height. For those who do not know these measurements, are provided with pictures to best help describe their body type.

Participants are also asked whether they've had a cough, fever, or change/loss in sense of smell/taste between January 2020 to March 2021 during registration. Participants are also asked whether they have had a swab prior to registration between January 2020 and March 2021, and whether this was positive, negative, unclear or they haven't received the results yet.

Our analyses of the primary outcomes will include the use of these infectious and non-infectious disease risk factors, and in particular issues with known associations such as BMI will be included. We will not be able to explore the stage of chronic disease (e.g. late stage vs early stage cancers or CVD) as we did not ask participants about this.

Reviewer comment 1-3:

Are there any intentions to stratify any of the analyses by virus strain? I understand that perhaps when the protocol was written, there was only one dominant circulating strain but as the protocol seems to be reactive to the changing epidemiology, it would be incredibly useful to include these data if possible.

Author response 1-3:

At present we don't think it will be possible to link Virus Watch samples to sequencing data, which would enable analyses by virus strain, but if this changes we would hope to undertake this work.

Reviewer comment 1-4:

Vaccination status is collected, hopefully the vaccine type along with the dates of vaccination are also recorded. Could vaccine efficacy also be estimated using these data? Duration of protection perhaps could also be estimated as there is a 5-year follow-up planned.

Author response 1-4:

Vaccine status is collected on a weekly basis. Individuals as part of the household are able to report whether they've received a vaccination. Included in this is which dose (1st or 2nd), who manufactured the vaccine (Pfizer/Bioentech, Oxford AstraZenica, Other, Don't know), and the date of the vaccine. We plan to use these data in analyses of vaccine effectiveness and we are hoping to receive permission to also link our data to national COVID-19 immunisation records.

Reviewer comment 1-5:

Certain behavioural interventions are hard to quantify, such as the extent of proper use of masks. How is this defined?

Author response 1-5:

We have now provided the following clarification around behavioural questions in the weekly survey ('Data collection and follow-up', paragraph 3):

"Questions around behavioural interventions, such as mask wearing and social distancing, aim to reflect the context and frequency/degree to which behaviours are practiced according to governmental and public health guidelines and relevant scientific literature."

We have also provided the following clarification for behavioural questions in the monthly surveys ('Data collection and follow-up', paragraph 4):

"As in the weekly questionnaire, questions around behavioural practices will reflect governmental and public health guidelines and the scientific literature; monthly questionnaires will also investigate barriers and enablers to health-related behaviours using purpose-developed questionnaires based on the Capability, Motivation, Opportunity, Behaviour (COM-B) model."

Reviewer comment 1-6:

One limitation is that only households with up to 6 persons can be included due to the REDCap survey infrastructure. This means that large multigenerational households will automatically be excluded and we know that this is an important risk factor for infection, particularly in BAME communities.

Author response 1-6:

We agree with the reviewer about this limitation and in the section on strengths and limitations (pg 4) we note this important limitation and another related limitation on the translation of our surveys:

- Only households with a lead householder able to speak English were able to take part in the study up until March 2021. From March 2021, translations of the online survey will be implemented for individuals recruited from this point onwards.
- Only households of up to six people were eligible for inclusion and they were also required to have access to an internet connection. These restrictions will limit the generalisability to large or multigenerational households, and those without access to the internet.

Reviewer comment 1-7:

The 5 year follow-up for indirect consequences of COVID-19 is particularly interesting, giving insights into how disruptions to the health system could impact longer-term health. This could potentially be one of the main aims as this study is uniquely placed to address this question.
Author response 1-7:

We agree with this comment and it will be an important aspect of our longer term research for the Virus Watch cohort.

Reviewer comment 1-8:

Given that we know that there is a significantly increased risk in both infection and disease severity in care home residents, has this been considered in either the recruitment or analysis plan?

Author response 1-8:

Virus Watch is set up to primarily study household transmission, and whilst we agree that increased risk in care home residents is important, the very large communal nature of these settings are very different to those we aim to investigate. Other studies, such as the Vivaldi study - <https://www.ucl.ac.uk/health-informatics/research/vivaldi-study> - are better placed to examine these risk factors and settings than Virus Watch.

Reviewer comment 1-9:

If self-administered nasal/throat swabs return inconclusive results (always a risk for example when trying to test children - notoriously difficult!), are follow-up tests offered?

Author response 1-9:

We are not asking for inconclusive results to be repeated at present because whilst we aim to process the swabs in as timely manner as possible, we are using postal services for the transport of swabs, and therefore there are delays between symptom onset and feeding back of results. Of the swabs processed so far, less than 3% have been inconclusive and therefore we do not think this will be a substantial problem with our data.

We have clarified in the manuscript, in the section on virus detection, that “We are not asking for inconclusive results to be repeated because we are unable to ensure these will be actioned and processed in a sufficiently timely manner for them to be appropriate.”

Reviewer comment 1-10:

Are data collected on whether children in the household are currently in school when samples / surveys are performed? This could be really useful for assessing the impact of school closures.

Author response 1-10:

We are not currently collecting this information, but will revisit this issue as we further develop our questionnaires.

Reviewer comment 1-11:

Has the trial been registered? I couldn't find the details.

Author response 1-11:

The study has recently been registered with ISRCTN:

Trial ID: ISRCTN32077121

Date registered: 12/03/2021

Link: <https://www.isrctn.com/ISRCTN32077121>

Reviewer comment 2-1:

Overall comment:

The current manuscript presents a study protocol to investigate incidence, symptom profiles, and transmission of COVID-19 in connection with population movement and behaviors at community level in the UK. The strength and limitations were raised by the team as indicated in Page 7. Overall, the manuscript is well written and organized, however, I suggest a number of comments to improve and make the manuscript more accessible.

Author response 2-1:

Thanks very much for this helpful review - we appreciate the time you have taken to carefully read and provide constructive feedback on the article.

Reviewer comment 2-2:

Specific comment:

Study design and setting

Can the team justify a scientific justification why they recruit 42,500 individuals including 12,500 minority backgrounds group? To estimate the number participants in any study, the team need to provide a specific information how they compute this estimate.

Author response 2-2:

We have clarified in the manuscript on page 9 that:

“Virus Watch is powered for our primary aims in study 2 and the estimation of population-level symptomatic COVID-19 attack rate over time. Recruiting a cohort that is representative of the population is time consuming as it requires an initial invitation into a study followed by multiple follow-up contacts encouraging invited individuals to register. Given the urgency of the public health situation to roll out our study as quickly as possible we chose a different approach whereby we recruit a large cohort of 45,000 individuals and from within that cohort we select a sub-sample for the testing cohort (sub-cohort 1) which is representative of the population in terms of area-level deprivation. The larger cohort will be important in assessing rates and predictors of less frequent outcomes such as hospitalisation and death. Given recent information of marked ethnicity differences in mortality rates from COVID-19 we also chose to recruit an ethnicity sample designed to be sufficiently large to provide early indicators of whether these differential mortality rates are due to differences in disease incidence or in differences in severity or both.”

And

“For the serology cohort of 3000 participants and assuming a modest design effect (DE) due to household and geographical clustering, 500 participants in six different minority ethnic groups would enable us to measure a cumulative incidence of 10% with 95% confidence intervals of +/-3% by group.

This sample size also provides 80 or 90% power to demonstrate statistically significant differences of 6.7% and 7.9% respectively between two groups.”

Reviewer comment 2-3:

Figure 1 simply presents an overview of cohort recruitment, and it has not directly associated with study design. Probably, the team need to revise the title of Figure 1.

Author response 2-3:

We have edited the title of Figure 1 to: “Overview of cohort recruitment, PCR swabbing schedules and data collection for the Virus Watch household community cohort study”

We believe this is an appropriate title as the figure illustrates how participants are recruited into studies 1 & 2, the swabbing protocols for study 2, and the different data collection methods including the geolocation tracking app and monthly surveys.

Reviewer comment 2-4:

Recruitment

To be enrolled in the study, participants require an internet connection using computer or mobile device. I understand this strategy will be reasonable and make the study accelerate under current circumstances we face now. However, this inclusion criteria may not be able to reach out a population who live in urban area or who has lower income, assuming that more likely such people are not accessible to internet connection. Thus, the study samples may not well represent the population the team is targeting. Can the team clarify this issue? Or any other strategy to have them in the study?

Author response 2-4:

We agree that requiring people to use an internet connection is a limitation of the study and is more likely to impact more deprived communities, however, we have had to balance this against the urgency of the study. We now note in the section on strengths and limitations:

“Only households of up to six people were eligible for inclusion and they were also required to have access to an internet connection. These restrictions will limit the generalisability to large or multigenerational households, and those without access to the internet.”

Reviewer comment 2-5:

I would think that the team needs to make a table for inclusion criteria and then put it in Appendix.

Author response 2-5:

We now list the full study inclusion and exclusion criteria in an appendix.

Reviewer comment 2-6:

It seems that the team already had an estimated clinical attack rate and its related quantities based on the previous data or their best knowledge. If it is, I encourage the team to write the paragraph (page 12) in the separate section, "Power Analysis". This should be an independent section to recruitment.

Author response 2-6:

We have now included a power analysis section below the recruitment section and included some additional details as to our approach to powering the study, as follows:

"Power Analysis:

The testing sub-cohort is powered for accurate weekly age-specific disease incidence rates to be measured assuming 20-30% clinical attack rate over 18 weeks. With a clinical Based on an estimated clinical attack rate of 30% of whom 20% need hospitalisation, and 0.5% die we expect the following number of outcome events in our testing cohort of 10,000 individuals in study 2: 3000 COVID-19 illnesses, 600 hospitalised cases, and 15 deaths. At one month into the outbreak we would be able to detect a 1.7-fold greater risk of disease in a population subgroup that constitutes 1/5 of the population, and by 2 months the detectable relative risk would be only 1.2. At one month we could detect a 4% hospital admission rate amongst cases with 95% CI of 0.5-6.8, and by 2 months the confidence intervals would narrow to 3.1-4.1. We have used estimates of the expected number of events over time to provide an indication of the fact that the cohort is sufficiently large to provide valuable information through the course of the pandemic. Sample size calculations have been informed by a realistic assessment of what we can achieve based on our previous experience[4,6]. For the serology cohort of 3000 people from minority ethnic backgrounds we assume a modest design effect (DE) due to household and geographical clustering, and 500 participants for six different minority ethnic backgrounds would enable the measurement of a cumulative incidence of 10% with 95% confidence intervals of 3% by each minority ethnic group. "

Reviewer comment 2-7:

Statistical Analysis

There is no clear analysis plan. What do the authors mean by using “appropriate regression models?” This section is an overview of the entire analysis in this study. It would be better for the readers to understand what analyses (names of analyses) will be used for in this Paragraph.

Author response 2-7:

We have added more detail on the types of models we plan to use in the main analyses listed (poisson regression with robust standard errors to account for clustering on the household level):

“Our primary analyses during the winter 2020/21 will focus on estimating age-specific weekly rates of symptoms and PCR-confirmed COVID-19 illness and hospitalisation. For this analysis we will use poisson regression models that account for clustering by household using robust standard errors and we will explore the use of stratification or weighting of the sample by age and region as necessary to give nationally representative estimates. Weekly rates will be expressed per 100,000 person-weeks for ease of comparison with national surveillance data.

We will examine the proportion of the population infected during the first wave (e.g. Feb 2020 to Sept 2020) and second and potentially future pandemic waves. We will estimate the percentage of the population infected by calculating age and wave-specific rates of serological infection and

PCR-confirmed disease per 100 person-seasons using poisson regression with robust standard errors to account for household-level clustering. A person-season will be defined by the epidemic curve in the cohort and therefore rates will account for differential follow-up time during each epidemic peak. In these analyses we will examine risk factors for infection, disease, disease severity and disease transmission.”

Reviewer comment 2-8:

On Study 1, the data will be collected from the online survey. Due to the nature of survey study, the team will face lots of missing values, eventually. Can the team clarify and include their strategies how to deal with such missing values (primary outcomes and predictors including demographic information) in the statistical analysis framework? This is very important issue in data collection from survey, and based on the study protocol, participants are being asked multiple times (every month) for different questions. So, this should be incorporated or at least mentioned in statistical analysis plan. In addition, I am also concerned how the team would check the quality of the data.

Author response 2-8:

We have added the following text under the title Missing data:

“We have several strategies that attempt to address the issue of missing data. First, we have sought to minimise the amount and impact of missing data for key outcomes and exposures through the study design. For example, a number of our primary outcomes (PCR+ illness, hospitalisation and death) and exposures (vaccination) we collect these both as self-reported data and through data linkage with the relevant national datasets and registries. Second, we sought to minimise missing serological and virus-watch specific swabbing outcomes in adults by making willingness to provide relevant specimens a prerequisite to study registration. Third, we know from our experience of previous community cohort studies of acute infections (Flu Watch and Bug Watch) that response to weekly surveys (where our symptom data is collected) is high at around 75%, which we believe is achieved by keeping these weekly data collections simple and quick to complete. We have aimed to replicate this approach in Virus Watch. Fourth, for important missing baseline demographic data (e.g. age and sex) we have created follow-up surveys to try and collect missing data at a later time in time. Fifth, where necessary, we will address missing data in our analyses and use multiple imputation methods if appropriate.”

Reviewer comment 2-9:

Modelling

I am confused upon reading this section, honestly. My understanding is that the team is trying to develop a new predictive modeling for a better prediction on different social distancing strategies. As the team described, they will be using the study data to train realtime disease prevalence estimation algorithms (need to cite relevant literature if needed), which indicates that the team needs to perform a machine learning approach. In this sense, it will be more appropriate to use “develop a predictive spatio-temporal transmission models” rather than “develop multi-level spatio-temporal transmission models”. I would like for the team to clarify this.

Author response 2-9:

This interpretation of the work is correct and an algorithm as described has been published in <https://www.nature.com/articles/s41746-021-00384-w> and the outcomes of this analysis are included in the weekly COVID-19 surveillance reports by PHE

(<https://www.gov.uk/government/publications/national-covid-19-surveillance-reports>). We have edited the description of the model as suggested to “predictive spatio-temporal transmission model”

Reviewer comment 3-1:

Overall comment: Major strength of the study is to take advantage of an already existing community cohort. However I believe the objectives are ambitious, and the methods are not

clear enough to understand if the objectives can be achieved through them, in particular objective 3 - "Virus Watch will measure effectiveness and impact of recommended COVID-19 control measures including testing, isolation, social distancing, respiratory and hand hygiene measures on risk of respiratory infection."

Author response 3-1:

Thanks very much for this helpful review - we appreciate the time you have taken to carefully read and provide constructive feedback on the article. We agree that our objectives are ambitious, but Virus Watch is a large collaborative study that builds upon our experience of running similar community cohort studies for the last 15 years. We have a wide range of disciplines involved including clinicians, public health, behavioural and computer scientists. Because of this depth and breadth of experience, we believe that whilst ambitious, we will be able to achieve these objectives and have been awarded competitive funding and ethical approval to carry this work out.

Reviewer comment 3-2:

Exclusion criteria seem to determine a major selection bias, that is not discussed in the limitations.

Author response 3-2:

We have updated the limitations of the study in the section of the protocol on strengths and limitations as per Author response 1-6 and 2-4.

Reviewer comment 3-3:

Finally, this protocol will need to be significantly amended to encompass effect of vaccination among the control measures. Again, for this objective the methods are not clearly described.

Author response 3-3:

Thank you for raising this important point. As noted in author response 1-4, we will ask participants regarding COVID-19 vaccine uptake on a weekly basis and we will also link to the national COVID-19 vaccination register in order to accurately determine vaccination status for all participants. We are also offering all adult participants (aged 18 years and over) monthly antibody testing on anti-Nucleocapsid and anti-Spike antibody assays from February 2021, which will allow us to assess immune responses to vaccination and detect asymptomatic infections following vaccination through seroconversion against Nucleocapsid. These data will allow us to study the impact of individual and household vaccination on SARS-CoV-2 infections. We will assess vaccine effectiveness against asymptomatic and symptomatic infections as one of our main outcomes, using two analytical approaches: a time-to-event analysis and a test-negative case-control analysis. The protocol has been amended throughout to include these points.

Reviewer comment 3-4:

A few specific comments below:

Introduction

In light of vaccine roll out, the introduction will need considerable review. Also, will effectiveness of vaccines be included as one of the control measures?

Author response 3-4:

Please refer to author response 3-3 where we hope we have now also responded to this issue.

Reviewer comment 3-5:

Study primary outcomes: will effectiveness of vaccine deployment be evaluated? If not, what is the plan to tease out the effect of the other measure, given vaccine roll out?

Author response 3-5:

We have added a new outcome "Vaccine effectiveness against asymptomatic and symptomatic infections." since submission of the protocol in Dec 2020, and we hope that this and our author response 3-3 address this comment.

Reviewer comment 3-6:

Recruitment

On "Households with more than six members will not be eligible for the study - this criteria was set due to limitations of the REDCap survey infrastructure which did not function correctly when attempting to work with household sizes of greater than six during our pilot testing of the survey.", I see a clear selection bias issue here. I wonder if study team would consider using a different software for the study instead of having this limitation as an exclusion criteria. Similar for the need to have internet and internet literacy to be included. Do authors have an idea of what proportion of the community stays out of the study with these exclusion criteria?

Author response 3-6:

We were required to use REDCap for this research because of information security and governance standards we had to meet. We explored alternative options to REDCap at the start of the project in April and May 2020, but were unable to find an alternative solution that met these requirements. Please also refer to author response 3-3 where we hope we have now also responded to this issue.

Reviewer comment 3-7:

Data collection and follow up

"6) Home antibody finger prick tests. 5000 members of the online cohort who are not part of the Laboratory testing sub-cohort (including 2500 minority ethnic and 2500 White British people) will be offered home finger prick antibody testing kits as soon as available after the first wave of the pandemic and after the second wave of the pandemic." - can the authors provide more information on tests to be deployed? And what defines "wave"? What if epidemiology changes and there is no clear "wave"? Would suggest defining time points for serial testing rather.

Author response 3-7:

We have expanded the antibody testing programme to offer all adult participants aged 18 years and over monthly antibody testing using at-home finger prick kits produced by Thriva Ltd. These CE-marked kits are designed for self-collection of a small volume (400-600 microlitres) of capillary blood using a retractable lancet. The kits, along with used lancets, are posted back to a laboratory where the samples are tested on validated anti-Nucleocapsid (semi-quantitative) and anti-Spike (quantitative)

electro-chemiluminescence immunoassays. Results will be sent to participants via email by the company providing the kits and antibody testing. These data will allow us to assess the antibody response to vaccination (anti-S), to detect asymptomatic seroconversions following vaccination (anti-N), and to measure antibody waning over time. Testing will commence in February 2021 and stop at the end of the study, which is currently August 2021. Kits will be posted to participants' homes on a monthly basis from the time of their enrolment into this aspect of the study. The protocol has been amended in several places to include these points.

VERSION 2 – REVIEW

REVIEWER	Mangal, Tara Imperial College London, Infectious Disease Epidemiology
REVIEW RETURNED	07-Apr-2021
GENERAL COMMENTS	Many thanks to the authors for addressing all points raised by the reviewers. I have no further comments.
REVIEWER	Gwon, Yeongjin University of Nebraska Medical Center, Biostatistics
REVIEW RETURNED	13-Apr-2021
GENERAL COMMENTS	No more comment.